# Impact of labor analgesia on mode of delivery and neonatal outcomes in Japan: A retrospective cohort study

**Shu Eguchi**[1]*, **Yuko Nagaoki**[1], **Sachiko Ohde**[2], **Michio Hirata**[1]

**1** Department of Pediatrics, St. Luke's International Hospital, Tokyo, Japan, **2** Graduate School of Public Health, St. Luke's International University, Tokyo, Japan

* ks0eh1nu0te17@gmail.com

## Abstract

Labor analgesia (LA) is associated with the potential hazard of high-risk delivery, such as cesarean section (CS) and instrumental vaginal delivery (IVD), and adverse neonatal outcomes such as neonatal asphyxia and respiratory distress. The objective was to examine the impact of LA on mode of delivery and neonatal outcomes and to counsel pregnant women about a potentially higher risk and allow them to decide LA or non-LA. This retrospective cohort study containing 5,184 pregnant women analyzed the association between LA and both mode of delivery and neonatal outcomes. LA increased the risk of IVD (Adjusted Odds Ratio [AOR] 3.25, 95% confidence interval [95%CI] 2.51–4.20) but decreased that of CS (AOR 0.52, 95%CI 0.44–0.60). Two factors (advanced maternal age [AOR 1.70, 95%CI 1.33–2.17] and primipara [AOR 4.72, 95%CI 3.30–6.75]) increased the risk of IVD. We should carefully consider the indication of LA for cases with these two factors since LA can increase the risk of IVD and adverse neonatal outcomes.

## Introduction

Childbirth, as one of the most important life events for most women, requires them to go through a series of burdens during labor and delivery. The magnitude of labor pain is often compared with a finger amputation [1]. Although the pain itself is not life-threatening for pregnant women and infants, it can transiently impair the cognitive function of pregnant women, especially memory function, during early puerperium [2]. Furthermore, the pain is related to the risk of postpartum depression, and the degree of pain is also associated with the severity of depression [3]. Researchers reported that analgesia can relieve these symptoms [4]. Currently, labor analgesia (LA) with various methods is commonly used in modern obstetric practice and includes spinal anesthesia, epidural analgesia, and combined spinal-epidural analgesia [5]. However, the side effects, especially those related to the mode of delivery and neonatal outcomes, are of great concern. Some studies reported that LA is associated with the potential hazards of cesarean section (CS), instrumental vaginal delivery (IVD), and adverse neonatal outcomes [6–12]. While many studies from Europe and the United States concluded that epidural analgesia does not increase the risk of CS or adverse neonatal outcomes [13–15], Seybe

of St. Luke's International University for researchers who meet the criteria for access to confidential data. The address for contact office of the Research Support Unit of St. Luke's International University is as follows e-mail: kenkyukikaku@luke.ac.jp.

**Funding:** Since this study was an observational retrospective study, written informed consent was waived and we proceeded using an opt-out approach. This study received no specific grant from any funding agency in the public, commercial, or not-for-profit sectors.

**Competing interests:** The authors have declared that no competing interests exist.

et al. indicated that epidural analgesia in Asians may increase the risk of emergency CS (ECS) with an adjusted odds ratio (AOR) of 2.35 (95% confidence interval [95%CI] 1.04–5.34) [6].

The risk of ECS after application of epidural analgesia may vary depending on race, and thus, a race-specific analysis should be carefully considered. In Japan, where the rate of LA was traditionally significantly low at 6% according to the Japan Society of Obstetrics and Gynecology, compared with 20%–70% in other countries in Europe and the United States, the focus has been on the application of LA in the past 5 years. It is concerning that too few reports on LA in Japanese and Asian pregnant women have been published [16, 17]. Specifically, advanced maternal age is so remarkable in Japan that the maternal age at first birth rose from 25.7 years of age (1975) to 30.7 years of age (2020), according to the Ministry of Health, Labour and Welfare, Japan, and this trend could naturally increase the risk of ECS. A meta-analysis reported by Pinheiro et al. showed that advanced maternal age itself was associated with an increased risk of induced labor and CS [18]. Therefore, it is urgent that data from Japan be analyzed due to the high proportion of advanced maternal age.

The research was whether LA has negative impact on mode of delivery and neonatal outcomes. We believe that this study, which includes an Asian population with a high rate of advanced maternal age, can contribute to the still controversial knowledge of the risks associated with LA. In addition, the present study aimed to counsel pregnant women about a potentially higher risk and allow them to decide LA or non-LA.

## Materials and methods

### Study population

We conducted a retrospective cohort study of all pregnant women who were admitted for delivery from April 2017 to August 2021 at St Luke's International Hospital, Tokyo, Japan, where the average number of deliveries is 1,400 per annum. We selected samples of pregnant women based on the inclusion and exclusion criteria shown in Fig 1. We included all pregnant women who were admitted for delivery in the study periods. The exclusion criteria were as follows Stillbirths and miscarriages were excluded. Since a mode of delivery was defined as the primary outcome, cases with elective CS were excluded. For the same reason, cases with multiple pregnancy and malpresentation that would have resulted in elective CS were also excluded. Cases of delivery at a gestational age of less than 36 weeks were excluded because these could affect a mode of delivery and neonatal outcomes. Finally, cases with elective LA, cases unavailable to LA, and cases with the lack of data due to outborn were excluded. All of the cases with elective LA had postoperative congenital heart disease, and the attending physician decided to conduct labor analgesia without the patient's wishes. The one patient with thrombocytopenia associated with Sjögren's syndrome who could not receive LA was excluded.

### Data collection

All data were obtained by chart review from the electronic medical records. Maternal data included age at delivery, which was dichotomized as advanced maternal age, defined as giving birth at age 35 or older [19], or not, height, pre-delivery body weight, pre-delivery body mass index, parity, duration of the second stage of labor, and whether or not LA was applied. Neonatal data included gestational age, gender, Apgar score (1 minute, 5 minute), admission to the neonatal intensive care unit (NICU), neonatal death, which is defined as death within 28 days of birth, and birth weight. The estimated fetal weight was replaced with the actual birth weight because the estimated fetal weight is not always accurate due to inter-examiner reliability. Pregnant women who received LA for vaginal delivery received either epidural analgesia, combined spinal-epidural analgesia, or dural puncture epidural.

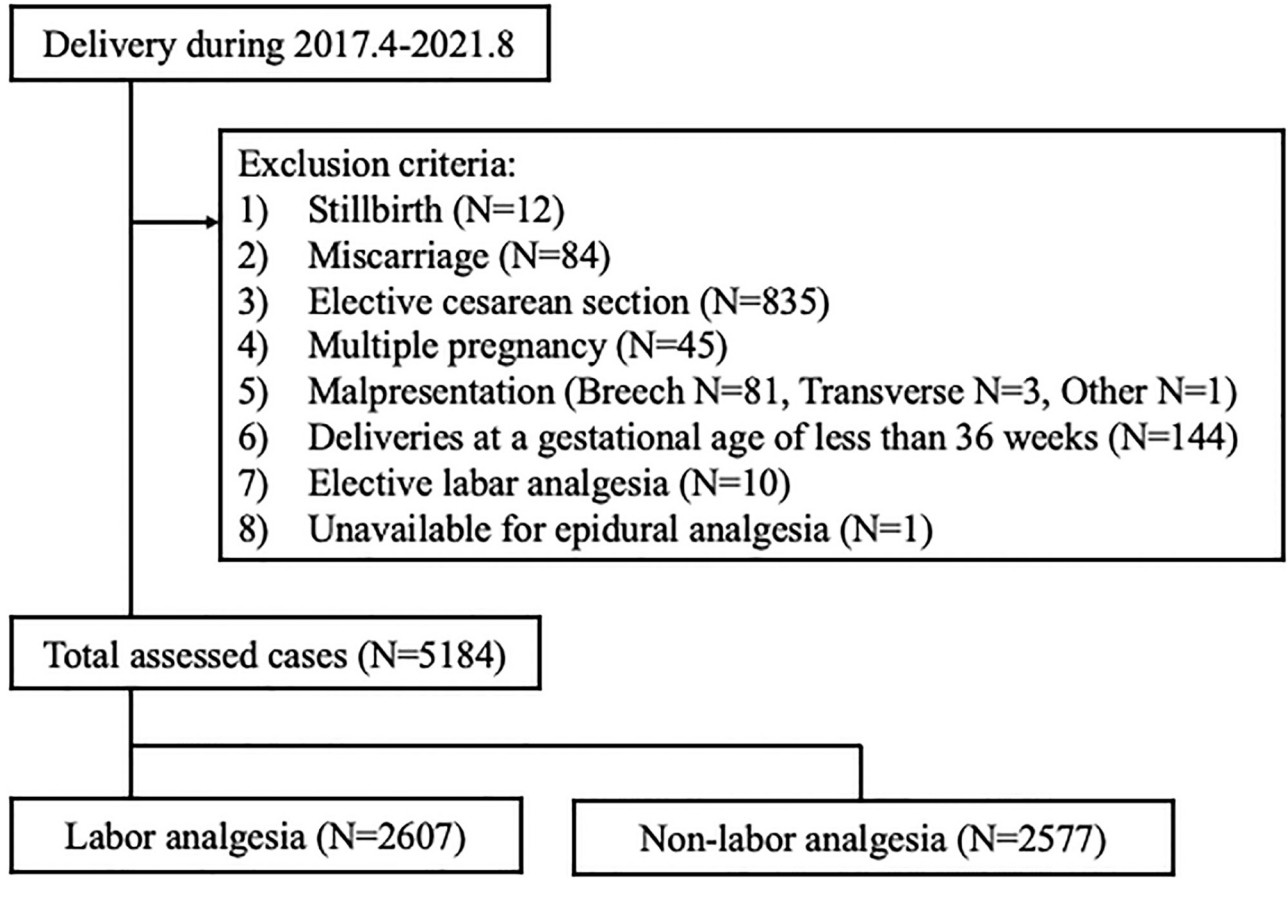

**Fig 1. The study patient flow diagram.**

## Hypothesis

First, we extracted the risks that may affect the mode of delivery and then examined the impact of LA on the mode of delivery. We hypothesized that the risks were LA, advanced maternal age, lower maternal height, larger estimated fetal weight, and primipara. The primary outcome of the present study was the mode of delivery, and the secondary outcomes were neonatal admission to the NICU and Apgar score. The indications for admission to the NICU were gestational age less than 36 weeks, birth weight less than 2,300 g, the occurrence of a respiratory disorder requiring supplemental oxygen or more, and when deemed necessary by neonatologists.

## Statistical analysis

When the association between mode of delivery and application of LA was examined, we performed the analysis using two different outcomes related to the mode of delivery: spontaneous vaginal delivery (SVD) and ECS ("SVD versus ECS") and SVD and IVD ("SVD versus IVD") with the reasoning that the mode of delivery is often automatically determined by the degree of the fetal head station. When immediate delivery is considered, ECS was selected in cases of high fetal head station, while IVD was selected in cases of low fetal head station. The high fetal head station means at +2 or more. The indications for the immediate delivery related to maternal factors included gestational hypertension, eclampsia, placental abruption

and HELLP syndrome. The indications related to fetal factors included fetal growth restriction and non-reassuring fetal status. The final decision is made by the attending obstetrician. Multivariate logistic regression analysis was performed to calculate the AOR and 95%CI, to examine the impact of LA on the mode of delivery and neonatal outcomes. In addition, we focused only on pregnant women who received LA and attempted to identify the characteristics of patients with factors that make LA risky. This study was approved by the institutional review board of our institute (Approval code: 21-R113) and was conducted according to the principles of the Declaration of Helsinki. Since this study was an observational retrospective study, the requirement for written informed consent was waived, which allowed us to proceed with this study in an opt-out manner. All data were analyzed using SPSS Statistics (IBM, Armonk, New York, USA). A result was considered statistically significant at $p < 0.05$.

## Results

### Patient characteristics

The total number of pregnant women who were admitted for delivery was 6,404. After considering inclusion and exclusion criteria, 5,184 women were finally included in the study population (Fig 1). The mean maternal age at delivery was 34.6 years with a standard deviation (SD) of 4.45 (Table 1).

**Table 1. Patients characteristics and association between labor analgesia group and Non-labor analgesia group at obstetrics inpatients.** (N = 5184).

| | All (N = 5184) | Labor analgesia group (N = 2607) | Non-labor analgesia group (N = 2577) | p-value |
|---|---|---|---|---|
| Age (year-old)[a] | 34.6 (4.45) | 34.7 (4.42) | 34.6 (4.49) | 0.4191 |
| Height (cm)[a] | 160 (5.37) | 160 (5.26) | 160 (5.47) | 1 |
| Pre-delivery body weight (kg)[a] | 52.4 (7.29) | 52.4 (7.12) | 52.5 (7.45) | 0.6213 |
| Pre-delivery BMI[a] | 20.5 (2.62) | 20.5 (2.56) | 20.5 (2.67) | 1 |
| Primiparous[b] | 3341 (64.4) | 1892 (72.6) | 1449 (56.2) | <0.001 |
| Multiparous[b] | 1843 (35.6) | 715 (27.4) | 1128 (43.8) | <0.001 |
| Duration of second stage of labor[a] | 12:33 (0:15) | 15:38 (0:46) | 9:04 (0:13) | <0.001 |
| Mode of delivery[b] | | | | |
| SVD | 3750 (72.3) | 1866 (71.6) | 1884 (73.1) | 0.218 |
| IVD | 412 (8.0) | 331 (12.7) | 81 (3.1) | <0.001 |
| ECS | 1022 (19.7) | 410 (15.7) | 612 (23.7) | <0.001 |
| With labor analgesia[b] | 2607 (50.3) | 2607 (100) | 0 (0.0) | 0 |
| Gestational age (weeks)[a] | 39.5 (1.2) | 39.6 (1.2) | 39.4 (1.3) | <0.001 |
| Male[b] | 2594 (50.0) | 1321 (50.7) | 1273 (49.4) | 0.359 |
| Birth weight (g)[a] | 3084 (377) | 3120 (364) | 3047 (387) | <0.001 |
| Apgar score[a] | | | | |
| 1 minute | 8.04 (0.936) | 7.96 (1.05) | 8.13 (0.797) | <0.001 |
| 5 minute | 9.03 (0.603) | 9.01 (0.628) | 9.05 (0.576) | 0.0169 |
| Admission to NICU[b] | 350 (6.8) | 172 (6.6) | 178 (6.9) | 0.654 |
| Neonatal death[b] | 0 (0.0) | 0 (0.0) | 0 (0.0) | |

[a]Mean (standard deviation), t-test p value.

[b]Number cases (percentage), chi-squared test p value.

BMI = Body mass index, ECS = Emergency cesarean section, IVD = Instrumental vaginal delivery, NICU = Neonatal intensive care unit, SVD = Spontaneous vaginal delivery

**Table 2. Association among mode of delivery at obstetrics inpatients.** (N = 5184).

| | SVD (N = 3750) | IVD (N = 412) | ECS (N = 1022) | p-value (SVD versus IVD) | p-value (SVD versus ECS) |
|---|---|---|---|---|---|
| Age (year-old)[a] | 34.1 (4.36) | 35.0 (4.54) | 36.0 (4.47) | <0.001 | <0.001 |
| Height (cm)[a] | 160 (5.27) | 160 (5.22) | 159 (5.67) | 1 | <0.001 |
| Pre-delivery body weight (kg)[a] | 52.2 (6.92) | 51.9 (6.99) | 53.5 (8.53) | 0.4041 | <0.001 |
| Pre-delivery BMI[a] | 20.4 (2.44) | 20.3 (2.50) | 21.2 (3.12) | 0.4309 | <0.001 |
| Primiparous[b] | 2112 (56.3) | 361 (87.6) | 868 (84.9) | <0.001 | <0.001 |
| Multiparous[b] | 1638 (43.7) | 51 (12.4) | 154 (15.1) | <0.001 | <0.001 |
| Duration of second stage of labor[a] | 11:33 (0:38) | 16:12 (0:20) | 17:04 (0:19) | <0.001 | <0.001 |
| With labor analgesia[b] | 1866 (50.0) | 331 (80.3) | 410 (40.1) | <0.001 | <0.001 |
| Gestational age (weeks)[a] | 39.4 (1.2) | 39.6 (1.2) | 39.5 (1.3) | 0.0013 | 0.0204 |
| Male[b] | 1797 (47.9) | 213 (51.7) | 584 (57.1) | 0.136 | <0.001 |
| Birth weight (g)[a] | 3084 (355) | 3084 (369) | 3078 (451) | 1 | 0.6525 |
| Apgar score[a] | | | | | |
| 1 minute | 8.22 (0.733) | 7.63 (1.09) | 7.56 (1.26) | <0.001 | <0.001 |
| 5 minute | 9.11 (0.525) | 8.85 (0.603) | 8.82 (0.781) | <0.001 | <0.001 |
| Admission to NICU[b] | 186 (5.0) | 33 (8.0) | 131 (12.8) | 0.008 | <0.001 |
| Neonatal death[b] | 0 (0.0) | 0 (0.0) | 0 (0.0) | | |

[a]Mean (standard deviation), t-test p value.

[b]Number cases (percentage), chi-squared test p value.

BMI = Body mass index, ECS = Emergency cesarean section, IVD = Instrumental vaginal delivery, NICU = Neonatal intensive care unit, SVD = Spontaneous vaginal delivery

The proportion of primipara was higher in the LA group (73%) than in the non-LA group (56%, p < 0.001) (Table 1). Primiparous women were more likely to receive LA (57%) than multiparous women (39%).

To examine the association between the mode of delivery and the application or not of LA, a univariate analysis was performed for IVD and ECS against SVD (Table 2). Compared with SVD, both IVD and ECS were associated with a higher proportion of admission to the NICU (IVD: 8.0%, p = 0.008 ECS: 12.8%, p < 0.001) and lower Apgar score (IVD 1 min/5 min: mean7.63/8.85, SD 1.09/0.603, p < 0.001/< 0.001 ECS 1 min/5 min: mean 7.56/8.82, SD 1.26/ 0.781, p < 0.001/< 0.001). The rate of LA was significantly higher in IVD (80%, p < 0.001) than SVD (50%) and conversely lower in ECS (40%, p < 0.001).

## Primary outcome: Impact of labor analgesia on mode of delivery

The primary outcome for mode of delivery is shown in Table 3. We analyzed two groups: one compared SVD and ECS, excluding IVD ("SVD versus ECS"), and the other compared SVD and IVD, excluding ECS ("SVD versus IVD").

In "SVD versus ECS," those of advanced maternal age and primipara exhibited a higher AOR (AOR 2.39, 6.00 95%CI 2.05–2.78, 4.95–7.28). Lower maternal height and larger esti- mated fetal weight increased the risks of ECS, while the use of LA decreased the risk of ECS (AOR 0.52, 95%CI 0.44–0.60).

Similar to the "SVD versus ECS" results, advanced maternal age and primipara also exhib- ited a higher AOR (AOR 1.66, 5.13 95%CI 1.33–2.05, 3.77–6.99) in the "SVD versus IVD" comparison. Maternal height and estimated fetal weight were not associated with the increased risk of IVD, but the use of LA increased the risk of IVD (AOR 3.25, 95%CI 2.51– 4.20).

**Table 3. The risk of labor analgesia against instrumental vaginal delivery and emergency cesarean section, and adverse neonatal outcomes at obstetrics inpatients.** (N = 5184).

| | Adjusted Odds Ratios | 95% Confidence interval | p-value |
|---|---|---|---|
| **ECS (versus SVD)** | | | |
| With labor analgesia | 0.52 | 0.44–0.60 | <0.001 |
| Advanced maternal age (≥ 35 year-old) | 2.39 | 2.05–2.78 | <0.001 |
| Height | 0.95 | 0.94–0.97 | <0.001 |
| Estimated fetal weight | 1.00 | 1.00–1.00 | 0.038 |
| Primiparous | 6.00 | 4.95–7.28 | <0.001 |
| **IVD (versus SVD)** | | | |
| With labor analgesia | 3.25 | 2.51–4.20 | <0.001 |
| Advanced maternal age (≥ 35 year-old) | 1.66 | 1.33–2.05 | <0.001 |
| Height | 0.99 | 0.97–1.01 | 0.386 |
| Estimated fetal weight | 1.00 | 1.00–1.00 | 0.741 |
| Primiparous | 5.13 | 3.77–6.99 | <0.001 |
| **IVD (versus SVD) among patients with labor analgesia** | | | |
| Advanced maternal age (≥ 35 year-old) | 1.70 | 1.33–2.17 | <0.001 |
| Height | 0.98 | 0.96–1.01 | 0.167 |
| Estimated fetal weight | 1.00 | 1.00–1.00 | 0.643 |
| Primiparous | 4.72 | 3.30–6.75 | <0.001 |
| **IVD (versus SVD) among patients without labor analgesia** | | | |
| Advanced maternal age (≥ 35 year-old) | 1.49 | 0.94–2.35 | 0.093 |
| Height | 1.02 | 0.97–1.06 | 0.481 |
| Estimated fetal weight | 1.00 | 1.00–1.00 | 0.895 |
| Primiparous | 6.23 | 3.37–11.52 | <0.001 |
| **Apgar score 1-minute <7** | | | |
| With labor analgesia | 2.06 | 1.50–2.83 | <0.001 |
| Advanced maternal age (≥ 35 year-old) | 1.29 | 0.96–1.74 | 0.088 |
| Height | 0.99 | 0.96–1.02 | 0.379 |
| Estimated fetal weight | 1.00 | 1.00–1.00 | 0.452 |
| Primiparous | 2.68 | 1.80–3.9 | <0.001 |
| **Apgar score 5-minute <7** | | | |
| With labor analgesia | 1.13 | 0.56–2.27 | 0.741 |
| Advanced maternal age (≥ 35 year-old) | 1.53 | 0.76–3.09 | 0.234 |
| Height | 0.92 | 0.86–0.99 | 0.015 |
| Estimated fetal weight | 1.00 | 1.00–1.00 | 0.227 |
| Primiparous | 2.07 | 0.88–4.87 | 0.096 |
| **Admission to NICU** | | | |
| With labor analgesia | 1.06 | 0.85–1.33 | 0.615 |
| Advanced maternal age (≥ 35 year-old) | 1.34 | 1.07–1.68 | 0.011 |
| Height | 1.01 | 0.99–1.03 | 0.338 |
| Estimated fetal weight | 1.00 | 1.00–1.00 | <0.001 |
| Primiparous | 1.68 | 1.30–2.18 | <0.001 |

ECS = Emergency cesarean section, IVD = Instrumental vaginal delivery, NICU = Neonatal intensive care unit, SVD = Spontaneous vaginal delivery

## Secondary outcome: Impact of labor analgesia on neonatal outcomes

The secondary outcomes, including neonatal outcomes and Apgar score, are shown in Table 3. LA lowered the Apgar score at 1 minute (AOR 2.06, 95%CI 1.50–2.83), but it did not influence

the Apgar score at 5 minutes or the NICU admission rate. Being a primipara was the strongest factor that could cause all adverse neonatal outcomes after adjusting for covariates including LA prescription.

When the population was restricted to only those who received LA, we found that pregnant women with the following two risk factors demonstrated a higher risk of IVD: advanced maternal age (AOR 1.70, 95%CI 1.33–2.17) and primipara (AOR 4.72, 95%CI 3.30–6.75) (Table 3). Neither lower maternal height nor larger estimated fetal weight were significantly associated with an increased risk of IVD.

## Discussion

This study included more than 5,000 pregnant women, approximately half of whom were of advanced maternal age, and who were admitted to our hospital for delivery from 2017 to 2021. In a multivariate analysis, LA increased the risk of IVD but decreased that of ECS. Moreover, advanced maternal age and primipara increased the risks of both ECS and IVD. Among those who received LA, two factors of advanced maternal age and primipara increased the risk of IVD; the AOR of primipara was highest at 4,72 (95%CI 3.30–6.75) and that of advanced maternal age was 1.70 (95%CI 1.33–2.17).

To the best of our knowledge, the present study is one of few reports from Japan that presents an association between LA and mode of delivery as well as neonatal outcomes. The novelty is that we sought to identify the characteristics of pregnant women who may need to be cautious in selecting LA. This study is valuable in that we analyzed primarily Asian patients, who are recognized to be at an increased risk of CS [6]; we also included a large number of patients of advanced maternal age.

As this results showed that LA decreased the risk of CS, many recent reports showed that LA is not associated with increased risk of CS or that LA even reduces CS rates [13, 15]. Despite a previous study published several decades ago, which concluded that epidural analgesia was associated with a significantly increased risk of CS among primipara [6], a recent Cochrane Review in 2018 that included 40 trials and more than 11,000 pregnant women found that epidural analgesia did not increase the risk of CS [14]. The reason that LA is not associated with and does not increase the ECS rate may be because factors associated with more painful labor are themselves associated with an increased risk of CS (e.g., fetal malrotation, fetal-pelvic disproportion, dysfunctional labor) [20]. Thus, we consider that pain relief might help to avoid ECS. However, no studies compared the degree of pain with the CS rate, and it is unclear why LA lowers the CS rate. Further investigation is needed to determine the pathophysiological reasons.

This results showed that LA also increases the risk of IVD. While some studies reported that LA is not associated with the increased risk of IVD [15], many other studies, including a systematic review by Lieberman et al. [8], reported the association between LA and an increased risk of IVD [7, 9, 10, 13]. Lieberman et al. estimated a pooled odds ratio of 2.2 (95% CI 1.3–7.8) [8], which is comparable to the result of the present study (AOR 3.25, 95%CI 2.51–4.20), and considered that the most common reason why pregnant women who receive LA are likely to undergo IVD is due to stalled labor, followed by risk to fetal well-being [8].

We found that the use of LA is associated with adverse neonatal outcomes. The effects of LA on neonatal outcomes are controversial and are the same as those on mode of delivery. Although a meta-analysis reported in the Cochrane Review in 2018 denied an association between epidural analgesia and adverse neonatal outcomes, such as admission rate to the NICU and an Apgar score of less than 7 at 5 minutes [14], many other studies confirmed the association [7, 11, 12]. In general, Apgar scores are lower in newborns delivered by CS [21,

22], but in this study, the Apgar score was lower in the LA group even though the CS rate decreased, which suggests that LA itself must lower the Apgar score. Kumar et al. reported that the Apgar score was significantly lower with use of epidural analgesia because of respiratory distress, and they considered that a large volume of distribution at a steady state in neonates resulted in a much longer terminal elimination half-life of fentanyl over several hours [11]. Most newborns with neonatal asphyxia are able to establish fetal-to-neonatal transition after appropriate resuscitation immediately after birth. This means that even if the Apgar score at 1 minute is low, the Apgar score at 5 minutes will be higher after appropriate resuscitation. This may be the reason why the results of the present study did not reveal statistically significant differences when the Apgar score at 5 minutes was less than 7. Researchers suggested that LA may elicit more severe adverse effects in a newborn in the immediate postnatal period than what the Apgar score at 1 minute purportedly represents. Neonatologists are expected to exhibit resuscitation skills for cases of neonatal asphyxia, especially in cases of LA. To promote resuscitation skills, we should provide a training and educational system to maintain higher skills and knowledge. Even if neonatal resuscitation is learned through simulation, this skill is difficult to maintain without practical experience, and thus, aggressive resuscitation practices are important. The centralization of facilities where LA is available might ensure safe deliveries. We should carefully consider LA, especially in facilities without personnel, such as pediatricians, neonatologists, and midwives, who attend deliveries.

According to the results, LA may reduce the risk of ECS but increase the risk of IVD and lower the Apgar score at 1 minute. In particular, the increased risk of IVD was significantly higher when the pregnant women who were primiparous and of advanced maternal age were administered LA. While the decrease in ECS is favorable in terms of surgical invasiveness and the safety of the next delivery, the increase in IVD is a serious issue. IVD is associated with many potential risks to both the mother (e.g., perineal and vaginal tears) and the newborn (e.g., brachial plexus injury with shoulder dystocia, subgaleal and intracranial hemorrhage) [23]. Prior et al. reported that the Apgar score was lower in infants delivered by IVD than in those delivered by CS [21]. Based on the results, we suggest that pregnant women at an increased risk of IVD, such as those of advanced maternal age (i.e., 35 years of age or older), lower maternal height, larger estimated fetal weight, and primipara, should carefully consider LA. Hence, when obstetricians and/or anesthesiologists confirm a pregnant woman's requests to receive LA, explaining not only the advantages but also the disadvantages and risks and obtaining appropriate informed consent is important. Moreover, pregnant women with advanced maternal age and/or primipara may be guided to receive alternative treatments, such as acupuncture, hypnotism, yoga, exercise during pregnancy, hydrotherapy, transcutaneous electronic nerve stimulation, massage, and relaxation techniques [5]. IVD techniques, especially obstetric forceps, are particularly dangerous in the hands of inexperienced obstetricians. Therefore, LA should be administered only in facilities with well-trained and experienced obstetricians because pregnant women who receive LA are more likely to require IVD.

The results of the present study are based on a limited population, and as mentioned above, we should be cautious about perceptions of the indications for and the risks of LA. Determining the indications for LA on a case-by-case basis and recognizing that LA is not beneficial for every pregnant woman is necessary because the population of pregnant women and facilities vary depending on the country, region, and other circumstances. The environment surrounding childbirth is changing; for example, advanced maternal age is increasing in all developed countries due to women's career advancement and lifestyle changes, and thus, we need to consistently update this research findings on the impacts of LA. In addition, the medical principle behind lower Apgar scores is not completely understood, and since the topic of LA remains controversial, further research is needed.

The present study also exhibits some limitations. The impact of the method of LA on the mode of delivery was not considered. In Tables 1 and 2, although statistically significant differences were observed in terms of maternal age, height, pre-delivery body weight, pre-delivery body mass index, gestational age, sex of newborn, and birth weight, these differences are not noteworthy in clinical practice because these factors were continuous variables and the number of cases that met the inclusion criteria was high. Similarly, in the multivariate analysis, we should focus on the interpretation of the comparison of the AOR of continuous variables, such as maternal height and estimated birth weight, and that of categorical variables, such as LA, advanced maternal age, and primipara. In addition, it is difficult to establish the cut-off for maternal height and estimated birth weight and decide clinical practice depend on the cut-off.

In conclusion, although LA is an important means to relieve pain during labor and delivery, it can increase the risk of IVD and an Apgar score of less than 7 at 1 minute. Therefore, we should carefully consider the indication of LA in cases of advanced maternal age and primipara. Furthermore, it is desirable to maintain and improve resuscitation skills through training and certification systems and to intensify facilities where LA is available.

## Supporting information

**S1 Checklist. STROBE statement—checklist of items that should be included in reports of observational studies.**
(DOCX)

## Author Contributions

**Conceptualization:** Shu Eguchi, Yuko Nagaoki, Michio Hirata.

**Data curation:** Shu Eguchi, Sachiko Ohde.

**Formal analysis:** Shu Eguchi, Sachiko Ohde.

**Investigation:** Shu Eguchi.

**Methodology:** Shu Eguchi, Yuko Nagaoki, Sachiko Ohde.

**Project administration:** Shu Eguchi.

**Supervision:** Sachiko Ohde, Michio Hirata.

**Writing – original draft:** Shu Eguchi.

**Writing – review & editing:** Shu Eguchi, Yuko Nagaoki, Sachiko Ohde, Michio Hirata.

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
