## [Decision Letter · Decision Letter 0]

27 Sep 2022

PONE-D-22-18968Impact of labor analgesia on mode of delivery and neonatal outcomes in JapanPLOS ONE

Dear Dr. Eguchi,

Thank you for submitting your manuscript to PLOS ONE. After careful consideration, we feel that it has merit but does not fully meet PLOS ONE’s publication criteria as it currently stands. Therefore, we invite you to submit a revised version of the manuscript that addresses the points raised during the review process.

We look forward to receiving your revised manuscript.

Kind regards,

Vicente Sperb Antonello, MD, MSc, Phd

Academic Editor

PLOS ONE

Journal Requirements:

Reviewers' comments:

Reviewer's Responses to Questions

**Comments to the Author**

1. Is the manuscript technically sound, and do the data support the conclusions?

Reviewer #1: Yes

Reviewer #2: Partly

Reviewer #3: Partly

Reviewer #4: Yes

Reviewer #5: Yes

Reviewer #6: Yes

2. Has the statistical analysis been performed appropriately and rigorously? 

Reviewer #1: I Don't Know

Reviewer #2: Yes

Reviewer #3: No

Reviewer #4: Yes

Reviewer #5: Yes

Reviewer #6: Yes

3. Have the authors made all data underlying the findings in their manuscript fully available?

Reviewer #1: No

Reviewer #2: No

Reviewer #3: No

Reviewer #4: Yes

Reviewer #5: Yes

Reviewer #6: Yes

4. Is the manuscript presented in an intelligible fashion and written in standard English?

Reviewer #1: Yes

Reviewer #2: Yes

Reviewer #3: Yes

Reviewer #4: Yes

Reviewer #5: Yes

Reviewer #6: No

5. Review Comments to the Author

Reviewer #1: This manuscript strongly supports the opinion that LA is not influenced to the increase of ECS. However, as the authors have mentioned in this article, the precise cut-off of maternal height or estimated fetal body weight have been still uncertain.

This time some concerns which must be discussed seem to remain in the manuscript. Please discuss or answer the following questions.

#1. Your institute often accept the foreign pregnant women from many countries. Is it necessary to discuss about the racial difference in this study?

#2. I would like to know that the precise indications for deciding IVD, which may help to clear the precise delivery outcome on LA.

#3. In Table 3, only the AOR and 95 % CI have shown in the IVD vs SVD with LA patients and suggested significant differences. However, the differences in estimated body weight among three groups (SVD, IVD and ECS) seem much closer and suggest no significant differences in Table 1. To avoid the misrecognition you should suggest more precise data in Table 3 or in the draft.

Reviewer #2: Dear Authors:

Although your manuscript is interesting and well-written, I think it does not have merit enough for publication in PLOS One, because it is only a retrospective cohort (please, include the study design in the title). There are lots of randomized controlled trials demontrating maternal and neonatal outcomes with labor analgesia and a Cochrane systematic review with more than 11,000 patients included. I suggest you try another journal and follow STROBE recommendations for observational studies, with a copy of STROBE checklist for cohort studies.

Reviewer #3: PONE-D-22-18968

Thank you for asking this important question. I have some concerns about the premise of the research, analysis, and conclusions.

Abstract:

1. What adverse neonatal outcomes are you specifically considering? It is mentioned that labor analgesia increases adverse neonatal outcomes, so perhaps mentions one or two examples.

2. The idea that the premise of this study was to identify people who should avoid labor analgesia seems problematic from the start. Perhaps the focus should be on counseling patients about a potentially higher risk of needing an operative vaginal delivery and allowing the patient to decide rather than advising them that they can’t get analgesia.

Introduction:

3. May be worth mentioning that labor pain is also associated with postpartum depression and anxiety.

4. I’m again troubled that the focus of this analysis is to identify people who should avoid labor analgesia. There are outcomes worse than requiting an operative delivery, and pain and depression may be among such outcomes.

Methods:

5. “The estimated fetal weight was substituted for the actual birth weight because the estimated fetal weight is not always accurate due to inter-examiner reliability.” Consider re-wording this as “The estimated fetal weight was replaced with the actual birth weight…”

6. Consider rewording this “When termination is considered,” as “When immediate delivery is considered” since termination has many meanings.

7. What are the indications for operative delivery at this institution? This may inform concerns related to this rate being higher. Many countries would interpret an increase in operative delivery over cesarean to be a success.

Results:

8. “When the population was restricted to only those who received LA, we found that pregnant women with the following four risk factors demonstrated a higher risk of IVD: advanced maternal age, lower maternal height, larger estimated fetal weight, and primipara (Table 3).” To be fair, you should do the same analysis on those who did NOT receive analgesia because you might find the same risk factors for operative delivery, so the analgesia may not have impacted this as much as you propose.

9. Indeed, for advanced maternal age and some other factors, analyzing only those with analgesia has a lower aOR than when you included everyone. Please also do the same analysis on those without analgesia.

10. The discussion mentions that operative delivery is associated with perineal tearing but this was not actually evaluated in this study. Please consider adding this analysis as it would help advise the results and discussion.

Discussion:

11. The aforementioned additional analysis on those who did NOT receive analgesia must be performed. These results should be reviewed after this sentence: “Among those who received LA, all four factors of advanced maternal age, lower maternal height, larger estimated fetal weight, and primipara increased the risk of IVD; the AOR of primipara was highest at 6.76 (95%CI 5.10–8.96) and that of advanced maternal age was second highest at 1.72 (95%CI 1.43–2.06).”

12. I remain unconvinced of the conclusions but await additional analyses.

Reviewer #4: The study entitled -Impact of labor analgesia on mode of delivery and neonatal outcomes in Japan - is interesting and evaluates the interference of labor analgesia on labor outcomes. This study included more than 5,000 pregnant women, which allows an interesting analysis of results.

However, some suggestions are made below:

In the Abstract

Adjusted Odds Ratio - put the confidence interval in the results. Introduction Last paragraph - just put the purpose of the study. It is not the place to describe the methodology.

In Materials and Methods

Describe inclusion and exclusion criteria. Figure 1 contains the exclusion criteria and does not have the inclusion ones. Both should be described in the methodology.

Lines 62 to 65 - the included text is disconnected from the rest.

Bibliographic references

Standardize the presentation of bibliographic references.

Reference 19 - has the year of publication in 2 locations.

As can be seen below: 19.Wong CA. (2014). Epidural and spinal analgesia/Anesthesia for labor and vaginal delivery, Obstetric Anesthesia: Principles and Practice. In: Chestnut DH (ed). Anesthesiology. Mosby: Maryland Heights, 2014, pp 490.

Reviewer #5: Labor analgesia (LA) is profit for pregnant women and those who are in labor, leading to a painless delivery. However, it is concerned in how to keep a lower risk (or better no risk) to neonatal and fetal health. This is a well-written manuscript analyzing the relationship between LA and possible risk factors. It will surely be helpful in providing new knowledge on obstetrics and in controlling and preventing adverse neonatal and fetal outcomes.

In general, the manuscript is fit for publication. In discussion section, data should be analyzed carefully, and conclusion should be made cautiously. Adverse neonatal and fetal outcome, cesarean section and instrumental vaginal delivery may be strongly associated with and even may be caused by advanced maternal age, lower maternal height, larger fetal weight, and primipara, and other risk factors, but not by LA. Perhaps, LA should not be restricted or prohibited in these pregnant women and those in labor with the four kinds of risk factors mentioned above. This should be re-analyzed carefully and added into the discussion section.

Reviewer #6: this manuscript is very important to be published. I suggested to revice minor as follow: Page 5 line 63 and 64 : sentence are not clear. It is better to write the criteria inclusion in narrative.

Page 8 line 105 = table title not com

plete should be completed with place, and when?, n=??

Page 9 line = 110-111 should be deleted, should not be written here.

Page 9 line 125 : table title not complete should be completed with place, and when?, n=??

Page 10 line 127-128 should be deleted, should not be written here.

Page11 line 137-138 table title not complete should be completed with place, and when?, n=??

Page 11 line 140 and page 12 line 141 should be deleted, should not be written here

Page 11 line 149 sentence is not clear

Page 17 Line 217-222; sentence and statement is not clear

6. PLOS authors have the option to publish the peer review history of their article (what does this mean?). If published, this will include your full peer review and any attached files.

Reviewer #1: No

Reviewer #2: No

Reviewer #3: No

Reviewer #4: No

Reviewer #5: **Yes: **Yi Lin

Reviewer #6: No

---

## [Author Response · Author response to Decision Letter 0]

1 Feb 2023

Dear Emily Chenette, Editor-in-Chief

I don't think I have deviated from the submission guideline, but I have received some pointers from Reviewer #6. I would appreciate any advice you could give me.

---

## [Decision Letter · Decision Letter 1]

8 Mar 2023

PONE-D-22-18968R1Impact of labor analgesia on mode of delivery and neonatal outcomes in Japan: A retrospective cohort studyPLOS ONE

Dear Dr. Eguchi,

Thank you for submitting your manuscript to PLOS ONE. After careful consideration, we feel that it has merit but does not fully meet PLOS ONE’s publication criteria as it currently stands. Therefore, we invite you to submit a revised version of the manuscript that addresses the points raised during the review process.

We look forward to receiving your revised manuscript.

Kind regards,

Vicente Sperb Antonello, MD, MSc, Phd

Academic Editor

PLOS ONE

Journal Requirements:

Additional Editor Comments (if provided):

I believe that the authors have made efforts to respond to reviewers' queries. Still, there are some points to be corrected. I suggest a new review and forward the article for further analysis.

Reviewers' comments:

Reviewer's Responses to Questions

**Comments to the Author**

1. If the authors have adequately addressed your comments raised in a previous round of review and you feel that this manuscript is now acceptable for publication, you may indicate that here to bypass the “Comments to the Author” section, enter your conflict of interest statement in the “Confidential to Editor” section, and submit your "Accept" recommendation.

Reviewer #1: All comments have been addressed

Reviewer #2: (No Response)

Reviewer #5: All comments have been addressed

Reviewer #6: All comments have been addressed

2. Is the manuscript technically sound, and do the data support the conclusions?

Reviewer #1: Yes

Reviewer #2: No

Reviewer #5: Yes

Reviewer #6: Yes

3. Has the statistical analysis been performed appropriately and rigorously? 

Reviewer #1: Yes

Reviewer #2: Yes

Reviewer #5: Yes

Reviewer #6: Yes

4. Have the authors made all data underlying the findings in their manuscript fully available?

Reviewer #1: No

Reviewer #2: Yes

Reviewer #5: Yes

Reviewer #6: Yes

5. Is the manuscript presented in an intelligible fashion and written in standard English?

Reviewer #1: Yes

Reviewer #2: Yes

Reviewer #5: Yes

Reviewer #6: Yes

6. Review Comments to the Author

Reviewer #1: Thank you for replying the reviewers’ comments.

Approximately a quarter of all cases have needed some interventions such as IVD or ECS regardless of LA in this study. The number of ECS was fewer in LA group compared with non-LA group. In contrast, the Apgar score at 1 min was lower in LA group. You already discussed about the discrepancy, such as the use of fentanyl, however, further analysis should be necessary.

As you mentioned, the indication of IVD or ECS was only determined by the fetal head station simply. I am confusing whether the fetal head station may influenced to the low Apgar score. Furthermore, we would like to know the reasons to decide the immediate delivery, for example, maternal factor or fetal factor. As you have abundant information over 5,000 valuable cases, more detail should be discussed as a final job in this study.

In addition, you should correct the sentence in L73. We may not be able to understand the meaning, ‘cases in which LA was unavailable for LA’.

Reviewer #2: I recognize the efforts of authors to do the modifications as indicated by other reviewers but I have the same concern I expressed before, and I don't think this retrospective study about labor analgesia in Japan has merit enough for publication in PLOS One.

Reviewer #5: The authors have addressed my comments properly by providing new Table 3. I think it is fit for publication.

Reviewer #6: Page 2 (Abstract) line please add recommendation and Keywords

Page 3 line 57 – 60 : The statement is not appropriate in the introduction. this has to be stated in the methodology .

Author should write the research question and the research's goal at the end of introduction section.

Page 7 line 81-82 : not clear why the criteria incusion : age older than 35 year??

Line 90: Outcomes and definition should be Hypothesis?

Page 10 line 119 : Table title should be completed: what, where??how many ?

Line 122-125 (page 11) : no need in this abbreviation under table (it is already in the methodology)

Page 11 line 137: Table title should be completed: what, where??how many ? and should not separated with the table (in the same page).

Page 12 line 139-142: no need in this abbreviation under table (it is already in the methodology)

Page 12 line 146-147 : This statement should be in the methodology as hypothesis of this study.

Page 13 line 151-152 : Table title is not complete (what, when, where and how many)

Page 14 line 154-155: no need in this abbreviation under table (it is already in the methodology)

Page 20 line 249-250 : what you mean pregnant women with these risk factors??

Page 20 Line 252-256 : the statement is not clear

Page 21 line 274-276: the statement is not clear

7. PLOS authors have the option to publish the peer review history of their article (what does this mean?). If published, this will include your full peer review and any attached files.

Reviewer #1: No

Reviewer #2: No

Reviewer #5: **Yes: **Yi Lin

Reviewer #6: No

---

## [Author Response · Author response to Decision Letter 1]

15 Mar 2023

Dear Emily Chenette, Editor-in-Chief of PLOS ONE

On behalf of all co-authors of this manuscript, I would like to thank the editor and the reviewers for a thorough review of our manuscript and constructive advice, which have improved the manuscript considerably. We carefully went over the important comments/suggestions from the reviewers and revised the manuscript. Please find our point-by-point responses to the reviewers’ comments below. All changes we made in this revision are highlighted in yellow. We hope that you will find the revisions and comments satisfactory.

Reviewer #1: 

#1. Approximately a quarter of all cases have needed some interventions such as IVD or ECS regardless of LA in this study. The number of ECS was fewer in LA group compared with non-LA group. In contrast, the Apgar score at 1 min was lower in LA group. You already discussed about the discrepancy, such as the use of fentanyl, however, further analysis should be necessary.

Response to the Reviewer:

We appreciate the reviewer’s peer review. Prior et al. reported that although the Apgar score was lower in infants delivered by IVD than in those delivered by CS, the reason is still unknown [22]. (Page 18, Line 240–241) Kumar et al. also reported that the effects of maternal epidural analgesia on neonatal respiration are not well described to date [11]. (Page 17, Line 215–218) In the present study, as well as other literatures, the discrepancy was unknown. (Page 17, 213–215)

As you mentioned, the indication of IVD or ECS was only determined by the fetal head station simply. I am confusing whether the fetal head station may influenced to the low Apgar score. Furthermore, we would like to know the reasons to decide the immediate delivery, for example, maternal factor or fetal factor. As you have abundant information over 5,000 valuable cases, more detail should be discussed as a final job in this study.

The indications for the immediate delivery related to maternal factors included gestational hypertension, eclampsia, placental abruption and HELLP syndrome. The indications related to fetal factors included fetal growth restriction and non-reassuring fetal status. The final decision was made by the obstetrician.

We followed the advice and add the above sentences. (Page 7, Line 106¬–109)

In addition, you should correct the sentence in L73. We may not be able to understand the meaning, ‘cases in which LA was unavailable for LA’.

We followed the advice and revised. (Page 5, Line 75)

Reviewer #6: 

Page 2 (Abstract) line please add recommendation and Keywords

Response to the Reviewer:

We appreciate the reviewer’s peer review. We followed the advice and add Keywords following the abstract. (Page 2, Line 27–28)

Page 3 line 57 – 60 : The statement is not appropriate in the introduction. this has to be stated in the methodology .

Author should write the research question and the research's goal at the end of introduction section.

We followed the advice and revised. (Page 4, Line 58–62)

Page 7 line 81-82 : not clear why the criteria inclusion : age older than 35 year??

The inclusion criteria did not contain maternal age. The data obtained included maternal age to dichotomize as advanced maternal age.

Line 90: Outcomes and definition should be Hypothesis?

We followed the advice and revised. (Page 6, Line 91)

Page 10 line 119 : Table title should be completed: what, where??how many ?

We followed the advice and revised. (Page 9, Line 125–126)

Table 1. Patients characteristics and association between Labor analgesia group and Non-labor analgesia group at obstetrics inpatients. (N=5184)

Line 122-125 (page 11) : no need in this abbreviation under table (it is already in the methodology)

We followed the advice and revised. The abbreviations were removed, but analysis methods were left as they are not described in the methodology.

Page 11 line 137: Table title should be completed: what, where??how many ? and should not separated with the table (in the same page).

We followed the advice and revised. (Page 11, Line 141)

Table 2. Association among mode of delivery at obstetrics inpatients. (N=5184)

Page 12 line 139-142: no need in this abbreviation under table (it is already in the methodology)

We followed the advice and revised. The abbreviations were removed, but analysis methods were left as they are not described in the methodology.

Page 12 line 146-147 : This statement should be in the methodology as hypothesis of this study.

We followed the advice and revised. (Page 6, Line 92–94)

Page 13 line 151-152 : Table title is not complete (what, when, where and how many)

We followed the advice and revised.

Table 3. The risk of labor analgesia against instrumental vaginal delivery and emergency cesarean section, and adverse neonatal outcomes at obstetrics inpatients. (N=5184)

Page 14 line 154-155: no need in this abbreviation under table (it is already in the methodology)

We followed the advice and revised. (Page 12, Line 150–151)

Page 20 line 249-250 : what you mean pregnant women with these risk factors??

We followed the advice and revised. (Page 19, Line 247)

Page 20 Line 252-256 : the statement is not clear

We followed the advice and revised. (Page 19, Line 249–250)

Page 21 line 274-276: the statement is not clear

We followed the advice and revised. (Page 20, Line 272–273)

In addition, it is difficult to establish the cut-off for maternal height and estimated birth weight and decide clinical practice depend on the cut-off.

Sincerely yours,

Shu Eguchi

Department of Pediatrics

St. Luke’s International Hospital

---

## [Editor Report · Decision Letter 2]

21 Mar 2023

PONE-D-22-18968R2Impact of labor analgesia on mode of delivery and neonatal outcomes in Japan: A retrospective cohort studyPLOS ONE

Dear Dr. Eguchi,

Thank you for submitting your manuscript to PLOS ONE. After careful consideration, we feel that it has merit but does not fully meet PLOS ONE’s publication criteria as it currently stands. Therefore, we invite you to submit a revised version of the manuscript that addresses the points raised during the review process.

After detailed analysis of the revisions, I consider that the authors made all the necessary modifications. Therefore, I consider that the article is eligible for publication in Plos One. Still, after my re-reading of the entire article, I suggest authors make the following changes before submitting the latest version:

1) Short title: Impact of labor analgesia on delivery and neonates. "I suggest: Impact of labor analgesia on delivery"

2) In the discussion section there are too many "our study, our results". It makes the reading a little bit tiresome. I would suggest use "In the present study..." for example.

3) I strongly suggest a full revision by a english native speaker.

We look forward to receiving your revised manuscript.

Kind regards,

Vicente Sperb Antonello, MD, MSc, Phd

Academic Editor

PLOS ONE
---

## [Author Response · Author response to Decision Letter 2]

27 Mar 2023

Dear Emily Chenette, Editor-in-Chief of PLOS ONE

On behalf of all co-authors of this manuscript, I would like to thank the editor and the reviewers for a thorough review of our manuscript and constructive advice, which have improved the manuscript considerably. We carefully went over the important comments/suggestions from the reviewers and revised the manuscript. Please find our point-by-point responses to the reviewers’ comments below. All changes we made in this revision are highlighted in yellow. We hope that you will find the revisions and comments satisfactory.

1) Short title: Impact of labor analgesia on delivery and neonates. "I suggest: Impact of labor analgesia on delivery"

Response to the Reviewer:

We appreciate the reviewer’s peer review. We followed the advice and revised. (Page 1, Line 3)

2) In the discussion section there are too many "our study, our results". It makes the reading a little bit tiresome. I would suggest use "In the present study..." for example.

We followed the advice and revised.

3) I strongly suggest a full revision by a english native speaker.

We followed the advice. The latest manuscript was revised by English native speakers.

Sincerely yours,

Shu Eguchi

Department of Pediatrics

St. Luke’s International Hospital

---

## [Editor Report · Decision Letter 3]

29 Mar 2023

Impact of labor analgesia on mode of delivery and neonatal outcomes in Japan: A retrospective cohort study

PONE-D-22-18968R3

Dear Dr. Eguchi,

We’re pleased to inform you that your manuscript has been judged scientifically suitable for publication and will be formally accepted for publication once it meets all outstanding technical requirements.

Kind regards,

Vicente Sperb Antonello, MD, MSc, Phd

Academic Editor

PLOS ONE

Additional Editor Comments (optional):

Regarding the last correction made by the authors, I believe that all questions have been resolved and the article can be forwarded for publication in Plos One.

---

## [Editor Report · Acceptance letter]

3 Apr 2023

PONE-D-22-18968R3 

Impact of labor analgesia on mode of delivery and neonatal outcomes in Japan: A retrospective cohort study. 

Dear Dr. Eguchi:

I'm pleased to inform you that your manuscript has been deemed suitable for publication in PLOS ONE. Congratulations! Your manuscript is now with our production department. 

Kind regards, 

on behalf of

Dr. Vicente Sperb Antonello 

Academic Editor

PLOS ONE